# Taking a snapshot of the triplet excited state of an OLED organometallic luminophore using X-rays

Grigory Smolentsev [1✉], Christopher J. Milne [1], Alexander Guda [2], Kristoffer Haldrup [3], Jakub Szlachetko [4], Nicolo Azzaroli [1], Claudio Cirelli[1], Gregor Knopp [1], Rok Bohinc[1], Samuel Menzi [1], Georgios Pamfilidis[1], Dardan Gashi[1], Martin Beck [1], Aldo Mozzanica[1], Daniel James[1], Camila Bacellar[1,5], Giulia F. Mancini[1,5], Andrei Tereshchenko[2], Victor Shapovalov[2], Wojciech M. Kwiatek [4], Joanna Czapla-Masztafiak[4], Andrea Cannizzo [6], Michela Gazzetto [6], Mathias Sander[7], Matteo Levantino [7], Victoria Kabanova[7], Elena Rychagova[8], Sergey Ketkov [8], Marian Olaru [9], Jens Beckmann [9] & Matthias Vogt [9,10✉]

OLED technology beyond small or expensive devices requires light-emitters, luminophores, based on earth-abundant elements. Understanding and experimental verification of charge transfer in luminophores are needed for this development. An organometallic multicore Cu complex comprising Cu–C and Cu–P bonds represents an underexplored type of luminophore. To investigate the charge transfer and structural rearrangements in this material, we apply complementary pump-probe X-ray techniques: absorption, emission, and scattering including pump-probe measurements at the X-ray free-electron laser SwissFEL. We find that the excitation leads to charge movement from C- and P- coordinated Cu sites and from the phosphorus atoms to phenyl rings; the Cu core slightly rearranges with 0.05 Å increase of the shortest Cu–Cu distance. The use of a Cu cluster bonded to the ligands through C and P atoms is an efficient way to keep structural rigidity of luminophores. Obtained data can be used to verify computational methods for the development of luminophores.

[1] Paul Scherrer Institute, 5232 Villigen, Switzerland. [2] The Smart Materials Research Institute, Southern Federal University, 344090 Rostov-on-Don, Russia. [3] Physics Department, Technical University of Denmark, DK-2800 Kongens Lyngby, Denmark. [4] Institute of Nuclear Physics, Polish Academy of Sciences, 31-342 Kraków, Poland. [5] Laboratory for Ultrafast Spectroscopy, Lausanne Center for Ultrafast Science (LACUS), École Polytechnique Fédérale de Lausanne, CH-1015 Lausanne, Switzerland. [6] Institute of Applied Physics, University of Bern, 3012 Bern, Switzerland. [7] ESRF, The European Synchrotron, 71 Avenue des Martyrs, 38000 Grenoble, France. [8] G. A. Razuvaev Institute of Organometallic Chemistry, Russian Academy of Sciences, Tropinina, 49, Nizhny Novgorod 603950, Russia. [9] Institute of Inorganic Chemistry and Crystallography, University of Bremen, Leobenerstr. 7, 28359 Bremen, Germany. [10] Present address: Martin-Luther-Universität Halle-Wittenberg Naturwissenschaftliche Fakultät II, Institut für Chemie, Anorganische Chemie, D-06120 Halle, Germany. ✉email: grigory.smolentsev@psi.ch; mavogt@uni-bremen.de

Organic light-emitting diode (OLED) technology is economical and powerful for the production of flexible displays and innovative area lighting[1–4]. Maximization of the fraction of gathered excitons used to produce light is a focal point in the development of high-performance electroluminescent devices. Classical organic dyes emit light due to fluorescence and have a theoretical limit of 25% for the internal quantum efficiency[4,5]. Coordination complexes encompassing heavy precious metals, such as Ir, Pt, and Ru, allow this limit to be overcome. Such dyes have strong emission from the lowest excited triplet state (so-called triplet-harvesting) and are commonly known as PHOLEDs (phosphorescent OLEDs). They can reach internal quantum efficiencies up to 100%[4–7]. However, the improved performance of PHOLEDs is connected to high costs because they require precious metals. In this regard, one of the main challenges is to develop efficient electroluminescent materials, which do not require rare and expensive transition metal ions, as this has a significant impact on the production cost of OLED devices and especially on possible applications for large scale area lighting[8,9]. During the last few years, a class of luminophores based on cost-efficient Cu coordination complexes has appeared, triggered by the discovery of the temperature-activated delayed fluorescence (TADF) effect[10–13]. A remarkable photoluminescence quantum yield >99% was recently achieved for such materials[14,15]. In parallel with the appearance of the first commercial OLED displays, we see now a revolution in the field.

A bright luminescence of Cu-based OLED materials is related to the specific properties of the triplet state[5]. Spin statistics govern an initial 1:3 ratio for electrically- generated singlet and triplet excitons. Therefore, if the spin–orbit coupling is small, as in classical organic materials, light is produced only as a result of the singlet excited state decay and the emission quantum yield is limited to 25%. For PHOLEDs the triplet state is emissive and a strong spin–orbit interaction enables the intersystem crossing from the excited singlet to the lower-lying triplet state. The approach for increased performance in Cu-based luminophores follows an orthogonal strategy, as in such coordination complexes

the spin–orbit coupling is sufficient to allow transitions between singlet and triplet states, but not strong enough for efficient emission from the triplet. Instead, such dyes are singlet-harvesters with a small singlet-triplet exchange energy (Fig. 1b). Triplet and excited singlet states are close enough that thermally activated back-transition from the triplet to the singlet (reverse intersystem crossing) can occur. Therefore, the triplet can store energy typically for microseconds, and this energy is subsequently released as delayed fluorescence following a thermally induced transition to the singlet state (this is the TADF effect)[16–20]. In this way, the internal quantum efficiency can also reach 100%.

There are two main aspects that influence the photoluminescence quantum yield of TADF materials: (i) the relative energy position of the lowest excited singlet and triplet states and (ii) the presence of non-radiative decay pathways from these states[11,17,21]. The first aspect influences the probability of the temperature-activated transition from the triplet to the excited singlet state. The required energy can be easily estimated from experimental steady-state and nanosecond emission measurements at different temperatures[11,16,17]. Regarding non-radiative decay channels they are typically temperature dependent and can be both intermolecular and intramolecular. Of particular noteworthiness, the most important intramolecular process for $Cu^I$ materials is the quenching of the excited state by vibrational coupling to the ground state[5]. In a simplified picture, intramolecular non-radiative decay becomes more probable if the equilibrium excited-state structure is displaced from the ground state along some vibrational coordinates[11] (Fig. 1d). A quantitative estimation of non-radiative processes requires advanced quantum calculations[22], which have to be verified experimentally with respect to the structural rearrangements and charge redistribution between the atoms of the complex. Such experimental verification can be obtained owing to the recent development of pump-probe techniques at X-ray free-electron lasers and synchrotrons.

X-ray absorption and emission spectroscopy are element-specific techniques, which are sensitive to the electronic structure (charge and spin state) of the probed chemical elements[23–25].

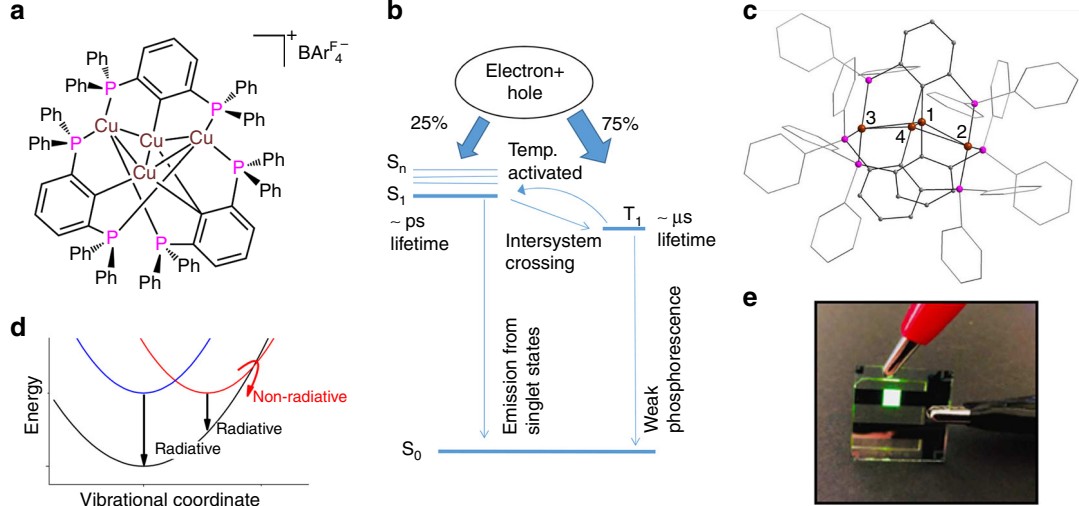

**Fig. 1 Temperature-activated delayed fluorescence: system and processes. a** $[Cu_4(PCP)_3]^+$ ($PCP = 2,6\text{-}(PPh_2)_2C_6H_3$) complex. **b** Scheme of light emission from Cu-based OLEDs due to temperature-activated delayed fluorescence (TADF) under electroluminescence conditions. After electron-hole recombination, the singlet and triplet excited states are occupied in a 1:3 ratio due to one possible momentum projection in the singlet and three possible projections in the triplet state. If the energy of the triplet state $T_1$ is close to the energy of the singlet $S_1$, temperature-activated reverse intersystem crossing ($T_1 \rightarrow S_1$) occurs which is followed by light emission from the singlet state. **c** Structure of $[Cu_4(PCP)_3]^+$ derived from single crystal X-ray diffraction. Cu atoms are brown, P atoms are magenta, C atoms are gray, H atoms are not shown. **d** Schematic illustration showing that non-radiative relaxation paths in Cu OLED materials are more probable if equilibrium excited and ground state structures are displaced along some vibrational coordinates. **e** Green emission from an OLED prototype with $[Cu_4(PCP)_3]^+$ as a luminophore.

To study charge-transfer processes, the measurements of absorption edges or emission lines of different elements are complementary because they give insight into the charge redistribution from the point of view of these atoms. Investigation of the excited state requires pump-probe versions of these techniques, which have been under active development during the last decade at synchrotrons and X-ray free-electron lasers[26–28]. Additionally, pump-probe X-ray scattering can provide information about the structural changes of the material, in particular about the relative displacements of heavy atoms, which dominate the X-ray scattering signal[29,30]. The combination of all these pump-probe X-ray methods in a single experiment is a strategic goal at the most advanced X-ray sources as they provide complementary information.

As a system for investigation, we have selected $[Cu_4(PCP)_3]$ $BAr^F_4$ ($PCP = 2,6\text{-}(Ph_2P)_2C_6H_3^-$, $Ar^F = 3,5\text{-}(F_3C)_2C_6H_3$) (Fig. 1a), a cationic organometallic $Cu_4$-cluster with a rigid ligand system involving Cu–Cu non-covalent interactions[31]. The optical properties of $[Cu_4(PCP)_3]^+$ are characterized by a bright green emission (with a maximum at 513 and 525 nm in the powder form and in tetrahydrofuran (THF) solution, respectively). The delayed photoluminescence lifetime in the solid state at room temperature is 9.8 μs. The photoluminescence efficiency is high in the solid state (up to 50%) and in frozen solution (up to 93%). $[Cu_4(PCP)_3]^+$ is a thermally robust material that does not show dynamic ligand exchange or solvent coordination. Its cationic nature and the selection of the appropriate counter anion allow for tunable solubility in organic media. This makes $[Cu_4(PCP)_3]^+$ compatible with solution-processed electroluminescent device production relevant for industrial applications. OLED prototype (Fig. 1e) that was solution-processed with $[Cu_4(PCP)_3]^+$ proved to be a potent light emitter[31]. Thus, $[Cu_4(PCP)_3]^+$ is a promising material for OLEDs and other electroluminescence devices.

$[Cu_4(PCP)_3]^+$ is a well-defined tetranuclear organo-copper cluster, in contrast to the vast majority of the copper complexes investigated for OLEDs which are either single or dinuclear coordination entities[11,16,21,32]. According to its ground state structure obtained with X-ray diffraction[31] (Fig. 1c), the four Cu centers are arranged in a concave kite-like rhombic structure with the angle between the two intramolecular planes defined by (Cu1–Cu2–Cu4) and (Cu1–Cu3–Cu4) of 29.3°. The diagonal Cu–Cu distances are 2.32 Å and 4.72 Å. There are two types of Cu arrangements in the system. Cu1 and Cu4 are coordinated by C atoms in pseudo-linear configuration with Cu–C distances of 1.92 and 2.05 Å. A distant carbanion is shared between two metals. Cu2 and Cu3 represent the second type of Cu centers: they are coordinated by three P-donors in a trigonal planar fashion with average Cu–P bond length of 2.27 Å. In this way, two CuP$_3$ units are capping a purely organometallic core. Thus, $[Cu_4(PCP)_3]^+$ exhibits a structure, which is reminiscent of a core-shell motif. This makes it an extraordinarily robust compound, even if Cu–C bonds are usually susceptible to hydrolysis and oxidation.

When considering the details of the electronic density and local structure changes of Cu-based TADF materials, the situation is clear for mononuclear complexes: metal to ligand charge transfer transitions occur and for efficient complexes these electronic transitions are accompanied by minimal structural rearrangements around Cu. Such conclusions were supported by theory and summarized in recent reviews[11,16]. For $[Cu_4(PCP)_3]^+$ only basic experimental data are available[31] and the theory describing the mechanism of luminescence requires experimental verification. The organo-multicopper cluster motif does not have well-studied analogues and it is not obvious how to transfer knowledge about the mechanism of luminescence of mononuclear Cu complexes to the multinuclear case. Therefore, the following three aspects are of key importance: (i) Which Cu atoms (C-coordinated or P-coordinated) are involved in the charge transfer? (ii) What is the role of the P atoms: do they only contribute to the structural integrity or do they also participate in the charge transfer? (iii) How do the distances between the Cu atoms change as a result of excitation?

To address these questions, we use a combination of time-resolved X-ray techniques: pump-probe X-ray absorption spectroscopy (XAS) at the *SLS* synchrotron, pump-probe X-ray emission spectroscopy (XES) at the *Swiss X-ray Free Electron Laser* (*SwissFEL*) and pump-probe X-ray solution-state scattering at the *ESRF* synchrotron. Exploring the high peak brightness of *SwissFEL* in the tender X-ray regime and state-of-the-art capabilities of pump-probe instruments at the *ESRF* and *SLS* synchrotrons we derive a complete picture of the charge transfer in the triplet excited state of this promising OLED material.

## Results

**Probing the charge of the Cu-cluster in the triplet state.** Time-resolved XAS spectra at the Cu K-edge were collected at the *SuperXAS* beamline of the *SLS*. The ground state and transient X-ray absorption near edge structure (XANES) spectra are shown in Fig. 2a. The setup for such experiments is working in the so-called pump-sequential-probes mode[33] and uses the synchrotron as a

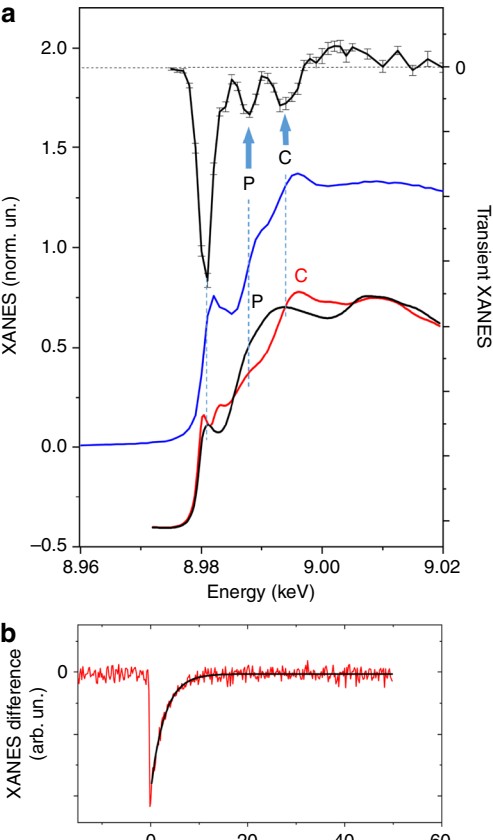

**Fig. 2 Pump-probe Cu K-edge X-ray absorption of $[Cu_4(PCP)_3]^+$. a** Experimental Cu K-edge transient X-ray absorption difference spectrum corresponding to the transition to the triplet excited state of $[Cu_4(PCP)_3]^+$ (1 μs time window after the photoexcitation) (black line with error bars (standard error of mean)), experimental ground state spectrum (blue line), theoretical ground state spectra corresponding to P-coordinated Cu atoms (black line) and C-coordinated Cu atoms (red line). **b** Kinetics measured using transient XAS for the incident beam energy 8.980 keV (red line) and exponential fit of these data (black line).

semi-continuous source: a pulsed laser excites the sample and a fast detection system measures the arrival time of all X-ray photons relative to the laser pulse with a time resolution of 30 ns. This approach allows for the simultaneous acquisition of the kinetics (Fig. 2b) and absorption spectra, which is optimal for experiments in the nanosecond-microsecond time range[34,35].

The absorption spectra at the Cu K-edge are sensitive to changes of the Cu oxidation state, but both non-equivalent Cu sites contribute to the measured XAS. These contributions can be separated if the kinetics of their response to the photoexcitation is different or by comparing them with theoretical calculations. For $[Cu_4(PCP)_3]^+$, only one transient species has been identified in the nanosecond-microsecond time range based on the decomposition of a series of 500 X-ray absorption spectra with principal component analysis (see Supplementary Fig. 1). A strong negative peak is present in the transient data at 8.980 keV, which is just below the ground state spectrum shoulder (Fig. 2a). Similar negative peaks have been reported in the literature and are typical signatures for oxidation at the Cu centers[36–40]. We observed that the temporal evolution of this feature is well described by a mono-exponential function with a time constant of 2.8 µs (Fig. 2b). Theoretical simulations of the ground state XANES (black and red curves in Fig. 2a) show that the main maximum of the spectrum corresponding to P-coordinated sites (at 8.992 keV) is shifted to lower energy relative to the maximum for C-coordinated Cu atoms (at 8.995 keV). Qualitatively, such shift is due to the influence of the first coordination sphere of Cu: for the quasi-linear coordination by two carbon atoms, the orbitals contributing to the main XANES maximum are non-bonding p-orbitals of Cu, while the coordination by three P atoms results in the hybridization of the P and Cu p- orbitals which shifts the main XANES maximum to lower energies. Oxidation of Cu influences the intensity of the shoulder at 8.980 keV and also shifts the maximum of the absorption spectrum to slightly higher energies. In the transient spectra, such shifts are seen as negative peaks at slightly lower energies than the main maximum. Therefore, if only P-coordinated sites change the oxidation state then one would expect a negative peak in the transient spectrum similar to that marked as P in Fig. 2a. If C-coordinated Cu sites are oxidized, then a negative peak similar to that marked as C in Fig. 2a would be expected. In the experimental transient data, we see both peaks P and C, which is an indication that both types of Cu atoms are involved in the charge transfer.

**Involvement of the phosphine ligands in the charge transfer.** Pump-probe X-ray emission spectra at the P Kα lines corresponding to the triplet excited state of $[Cu_4(PCP)_3]^+$ were measured as one of the pilot experiments at *SwissFEL*. The scheme of the experimental setup is shown in Fig. 3a. In contrast to synchrotrons, *SwissFEL* is a low repetition rate facility (10 Hz in our experiment) which provides a high number of photons in a single, ultrashort X-ray pulse ($\sim 3 \times 10^{11}$ photons/pulse with a duration of 50–100 fs). Pump-probe X-ray emission spectroscopy is a technique requiring a high photon flux and efficient photo-excitation, conditions that can be more easily achieved at the repetition rate of X-ray Free Electron Lasers, XFELs (typically, below 1 kHz) rather than at synchrotrons. Indeed, a lot of XES experiments in the hard X-ray range have been performed at XFELs[41–44]. However, the tender X-ray range (1–5 keV) is challenging for most of the X-ray facilities because it lies in between ranges accessible with soft and hard X-ray optics, making it a technically difficult spectral region to cover. In this regard, *SwissFEL*[45] provides unique possibilities for spectroscopy allowing experiments at the K-edges of S, P, Cl, Ca as well as the L-edges of the 4d metals (Ru, Rh, Pd, Ag, etc.). Non-resonant P Kα

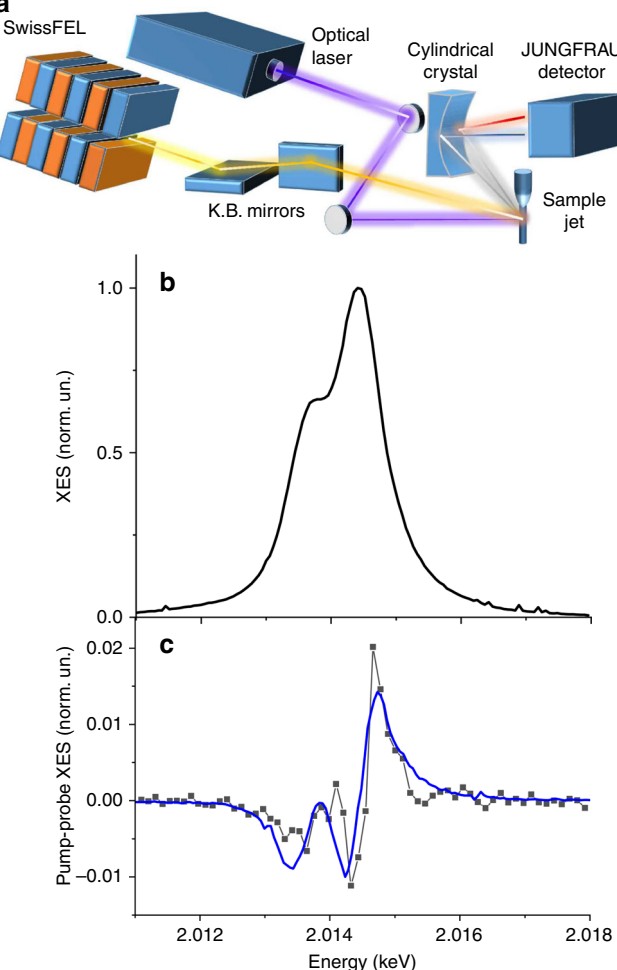

**Fig. 3 Pump-probe P Kα X-ray emission spectroscopy for $[Cu_4(PCP)_3]^+$. a** Scheme of the pump-probe P Kα X-ray emission experiment at *SwissFEL*. The X-ray beam from *SwissFEL* is focused with *Kirkpatrick–Baez* (K.B.) mirrors and interacts with the sample jet. The same sample volume is excited by an optical laser. X-ray fluorescence from the sample is dispersed using a cylindrically bent crystal (*von Hamos* type geometry) in the horizontal plane and measured using a 2D JUNGFRAU detector. **b** Ground state P Kα X-ray emission spectrum of $[Cu_4(PCP)_3]^+$. **c** Pump-probe P Kα XES signal (black line) corresponding to the triplet excited state of $[Cu_4(PCP)_3]^+$ and the signal calculated from the expected shift of emission lines (blue line).

XES spectra were measured using a *von Hamos* spectrometer that disperses X-rays in the horizontal plane and focuses them vertically. In this configuration the point of interaction of X-rays with the sample, cylindrical crystal, and JUNGFRAU detector are located in the same horizontal plane. This type of spectrometer allows for the full X-ray emission spectrum to be measured on a shot-to-shot basis, which is the most efficient strategy for X-ray sources with fluctuating X-ray intensity such as XFELs[46] since it collects a full spectrum for every XFEL pulse, allowing the spectra to be sorted and filtered on a shot-to-shot basis. For measurements such as this, where the data were collected non-resonantly with no scanning elements, the spectral shape of XES is independent on the incident X-ray intensity and the spectra can be simply summed together for efficient data collection.

The X-ray emission spectra around the P Kα lines are sensitive to the electronic density of this chemical element and therefore pump-probe XES allows for evaluation of the involvement of the P atoms in the charge transfer. The experimental ground state

spectrum is shown in Fig. 3b and the transient XES, corresponding to the triplet state, in Fig. 3c, black line. The transient has been extracted using principal component analysis from the measurements for delays varied in the range 1.4 ns. Two lines ($K\alpha_1$ at 2014.4 eV and $K\alpha_2$ at 2013.6 eV), which form the ground state spectrum, shift to higher energies with the increase of P charge. This leads to the XES difference spectrum with one positive and two negative peaks (Fig. 3c, blue line). From measurements on reference compounds[47], the expected shift of the main line is 0.1 eV per 1.0 electron change of the formal P charge (oxidation state). Alternatively, one can use a calibration to the density functional theory (DFT) charge[48], for example, calculated using Mulliken approach. In this case, the same 0.1 eV shift of the emission line corresponds to the 0.13 electrons of the DFT charge variation. From the comparison of the amplitude of transient XES with spectral changes induced by such shift, we can estimate that the change of the average charge of P atoms is 0.097 electrons of formal charge or 0.013 electrons of DFT charges (see Supplementary Methods for more details). The magnitude of the transient signal is also directly proportional to the excited state fraction, which we have maximized, but we do not expect that it exceeds 70%, which is the maximum reported in the literature for pump-probe XAS experiments[49]. Thus, our estimate of the formal charge change by at least 0.097 electrons should be considered as a lower bound for this parameter.

**Probing the structure of the triplet state of $[Cu_4(PCP)_3]^+$.** Pump-probe X-ray solution-state scattering (also known as wide-angle X-ray scattering, WAXS) measurements have been performed at the *ID09* beamline of the *ESRF* synchrotron. In this setup, a fast mechanical X-ray chopper is used to isolate individual X-ray pulses from the synchrotron at a 1 kHz repetition rate. Synchronization of the arrival time of optical and X-ray pulses allows probing the sample at different times following photoexcitation. X-ray scattering patterns are collected with a 2D detector placed behind the liquid-jet sample.

X-ray scattering is particularly sensitive to the structural changes involving the most electron-rich atoms of the system, therefore, the relative displacements of the Cu atoms can be directly probed using this technique. Experimental time-resolved X-ray scattering patterns at 100 ps, 1 ns, and 2 μs delays from laser photoexcitation, which were obtained after azimuthal integration of the signal from the 2D detector, are shown in Fig. 4 (black lines). The pump-probe signals mainly arise from the rearrangement of the $[Cu_4(PCP)_3]^+$ structure and from the heating and density changes of the bulk solvent. The solvent response (see Supplementary Fig. 2) was measured in the reference experiments using the same approach as in a previous report[50]. They show a characteristic difference signal with sharp features in the Q range 0.7–1.6 Å$^{-1}$. The solvent contribution dominates the observed difference signal for the long time delay (2 μs), but has a comparable amplitude to the solute contribution for short delays (100 ps, 1 ns). The contribution of $[Cu_4(PCP)_3]^+$ to the total pump-probe X-ray scattering is seen as oscillations in the range 1.5–6 Å$^{-1}$ (Fig. 4, red line). We have calculated the scattering signals for the DFT-based models of the ground and excited triplet states reported[31]. A linear combination of such theoretical data and solvent signal (Fig. 4, blue line) agrees well with the experimental data measured at 1 ns after excitation. In these structures, the distance between the C-coordinated Cu atoms increases by 0.05 Å and between the P- coordinated atoms decreases by 0.12 Å as a result of photoexcitation. The average distance between C-coordinated and P-coordinated Cu atoms changes from 2.87 Å in the singlet state to 2.83 Å in the triplet.

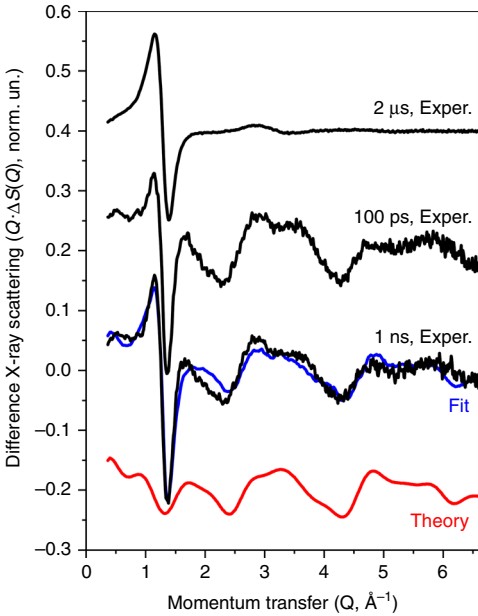

**Fig. 4 Pump-probe X-ray scattering for $[Cu_4(PCP)_3]^+$.** Experimental pump-probe X-ray scattering signals corresponding to a 2 μs, 100 ps, and 1 ns delay from photoexcitation (black lines). Theoretical X-ray scattering difference for the DFT-based models calculated taking into account structural changes of $[Cu_4(PCP)_3]^+$ as a result of the transition to the triplet state (red line). Fit that takes into account additionally the bulk solvent response due to ultrafast heating (blue line). The signal corresponding to 2 μs delay has been divided by 6.5 to match the scale.

## Discussion

The present study demonstrates how three different pump-probe X-ray techniques provide complementary insights. Element-selective information about the charge at Cu and P atoms obtained by time-resolved spectroscopic methods was complemented by the structural information from X-ray scattering. One of the trends that are pursued at an advanced X-ray facilities is to combine pump-probe spectroscopy and scattering techniques in one experiment[41,42,51]. We have selected another approach with three experiments performed at three different facilities. X-ray emission and scattering measurements often can be combined, but not in the case of XES in the tender X-ray regime. In some cases, X-ray emission can be used instead of X-ray absorption to monitor the electronic structure changes and Cu XES and X-ray scattering are technically possible combinations. In our case, the measurements of the Cu $K\alpha_{1,2}$ or $K\beta_{1,3}$ emission are not favorable due to low sensitivity of these spectra to the oxidation state[52] (in comparison to, for example, Fe[23]) while XAS has high sensitivity. Thus, we see complementarity at the level of pump-probe techniques and also at the level of user facilities for X-ray experiments.

Combining the data obtained with all three X-ray methods, the following picture emerges: for the triplet excited state the charge shifts from the orbital delocalized on both P-coordinated and C-coordinated Cu atoms and involving also the Cu-bound phosphorus atoms. This suggests that the charge moves to the phenyl rings, which is in agreement with DFT calculations which predicted more negative charges on the C atom of the bridging phenyl ring bound to Cu1 and Cu4 in the triplet state[31]. The electron transfer is accompanied by a structural re-arrangement involving an increase of the distance between C-coordinated Cu atoms by 0.05 Å and a decrease of the average distance between C- and P- coordinated Cu atoms by 0.04 Å.

**Table. 1 Differences between average atomic charges for excited triplet state and ground state. .**

| Code | Level of theory | Charge analysis method | Δq for Cu (P-coord)[a] | Δq for Cu (C-coord)[a] | Δq for P[a] |
|---|---|---|---|---|---|
| Gaussian | B3LYP/6-311G(d,p) | NBO[b] | −0.002 | 0.203 | −0.004 |
| Gaussian | B3LYP/DGDZVP | NBO[b] | −0.049 | 0.080 | 0.013 |
| ADF | TPSS/QZ4P | Mulliken[b] | 0.099 | −0.066 | 0.010 |
| ADF | B3LYP/QZ4P | Mulliken[b] | 0.078 | 0.067 | −0.017 |
| ADF | B3LYP*/TZ2P | Bader[b] | −0.019 | 0.082 | 0.020 |

[a]Average change of charge (triplet–singlet state) are reported separately for P-coordinated Cu atoms, C-coordinated Cu atoms, and P atoms.
[b]Charges were calculated using Mulliken, natural bond orbital (NBO)[67] and Bader[68,69] approaches.

To compare our experimental results about charge shifts with theoretical data we have summarized in Table 1 a few representative examples of DFT caclulations obtained using the Gaussian and ADF codes. Average changes of the charges are given in this table for P atoms, C-coordinated Cu and P-coordinated Cu atoms. An extended version of this table (Supplementary Table 3), as well as individual charges for all Cu and P atoms calculated using different methods (Supplementary Tables 1 and 2, Supplementary Fig. 5), are provided in the supplementary information. From Table 1 one can give a broad range of interpretations. From the NBO charges obtained with Gaussian at the B3LYP/6-311G(d,p) level of theory one can conclude that the electron transfer occurs only from C- coordinated Cu sites. Using the same code, but with a smaller basis set (B3LYP/DGDZVP) one can deduce that the negative charge moves from the C-coordinated to P-coordinated Cu atoms. ADF at TPSS/QZ4P level of theory gives an opposite result: the electron density moves from the P-coordinated to C-coordinated Cu atoms. Using ADF (B3LYP/ QZ4P) and Mulliken charges we see the shift of the electronic density from both C coordinated and P- coordinated Cu atoms, which is in agreement with the qualitative picture obtained from the pump-probe XANES. The Bader charges computed at the B3LYP*/TZ2P level of theory testify for a significant contribution of P atoms in addition to the C-coordinated Cu atoms. In the set of calculations that we have performed, we have not found the approach that shows the electronic density shift from both types of Cu and P atoms. The charge change at P atoms (0.020 e) obtained using ADF (B3LYP*/TZ2P) is in agreement with our experiment-based low estimate for the DFT charge change (0.013 e). Thus, a broad range of theories about charge shift in $[Cu_4(PCP)_3]^+$ as a result of photoexcitation demonstrates the need of experimental verification based on pump-probe X-ray measurements.

Some computational models, for example, Mulliken charges calculated using Gaussian within B3LYP/6-311G(d,p) approach, shows that the charge moves mainly from one C-coordinated Cu atom. Our experimental results demonstrate the involvment of two types of Cu centers. In the excited state, it is advantageous to prevent the formation of a four-coordinate $Cu^{II}$ with relatively flexible coordination sphere. $Cu^{II}$ prefers a square planar geometry. Therefore, ligands often rearrange in such excited state so that the metal interacts with the solvent. This effect is exciplex formation and it was observed previously for Cu complexes[38,53]. It leads to the quenching of the excited state because significant energy is required for the structural rearrangement. The clustered structure triggered by the *PCP* ligands is rather rigid and prevents distortions towards four-coordinated planar $Cu^{II}$. The probability of non-radiative losses depends on the coupling of vibrational modes of the ground and excited states, which can be described using *Huang–Rhys* parameters. For rigid complexes that demonstrate small structural difference between ground and excited state *Huang–Rhys* parameters are minimal. Our data demonstrate that such a situation occurs for $[Cu_4(PCP)_3]^+$.

The strategy of minimizing non-radiative losses by reducing *Huang-Rhys* parameters using tridentate ligands has been previously explored, for example, for Pt-based complexes utilized in OLED devices.[5,54] The multicore approach realized in $[Cu_4(PCP)_3]^+$, complements other strategies of keeping structural rigidity and lowering the reorganization energy such as implementing steric hindrance in mononuclear $Cu^I$ complexes with tetrahedral coordination (thus hampering a flattening of the structure upon photo-induced oxidation to $Cu^{II}$) or reducing the coordination number of Cu to two or three[11,16,17]. Further development of such complexes can be based on the alteration of the P-donors and phenyl-backbone to tune the emission maximum. Since both Cu and P atoms are involved in the charge transfer in the excited state it is worth to consider modifications of ligands to change independently the electronic density at P an Cu centers and to explore their influence on the excited state and luminescence properties. Computational methods with experimentally verified parameters can be of great help predicting optimal ligands that correspond to the most efficient TADF emitters. The delocalized charge movement and structural rigidity of $[Cu_4(PCP)_3]^+$ combined with high solubility and stability allow us to conclude that cationic organometallic copper clusters represent a family of promising materials for optoelectronic devices.

## Methods

**Pump-probe XANES measurements and analysis**. Pump-probe XANES measurements at the Cu K-edge have been performed at the *SuperXAS* beamline of the *Swiss Light Source* (*SLS*, Villigen, Switzerland). The storage ring was run in the top-up mode with an average current of 400 mA. The pump-sequential-probes XAS setup acquired data in the asynchronous mode[33]. The X-ray beam was collimated by a Si-coated mirror at an incidence angle of 2.5 mrad, which also served for harmonic rejection. The energy has been scanned by a channel-cut Si(111) monochromator. A toroidal mirror with Rh coating was employed after the monochromator to focus the incident X-rays to a spot size of $100 \times 100 \ \mu m^2$. The photon flux at the sample was about $4 \times 10^{11}$ photons/s. The excitation with 447 nm wavelength was provided by a Xiton IDOL laser with a repetition rate of 5 kHz, a pulse duration of 12 ns and output power of 0.2 W. The laser beam fluence of ~130 mJ cm$^{-2}$ was achieved by focusing to the $200 \times 200 \ \mu m^2$ spot at the sample position. Approximately 150 mL of sample were circulated in the closed cycle flow system with laser and X-ray beams focused on the round jet of the sample with a diameter of 750 μm. This jet was placed in a chamber filled with $N_2$. The concentration of $[Cu_4(PCP)_3]BAr^F_4$ in the anhydrous THF solution was 2 mM. The solution was purged with $N_2$ for 30 min before optical and X-ray irradiation to remove dissolved oxygen and was kept in $N_2$ atmosphere and continuously purged during the measurements.

Theoretical XANES spectra have been obtained by calculating the probabilities of transitions between the core and virtual molecular orbitals using previously reported method[55,56]. Molecular orbitals were obtained by DFT using ADF code[57]. Non-relativistic self-consistent calculations have been performed using the quadruple-ζ basis set with four polarization functions (QZ4P) and the hybrid B3LYP exchange-correlation functional. Convolution of spectra has been made within the arctangent model, which takes into account contributions from the core-hole lifetime broadening, experimental resolution, as well as the energy-dependent broadening due to the finite mean free path of the photoelectron.

**Pump-probe XES measurements and analysis**. The pump-probe X-ray emission spectra around the P Kα lines have been measured as a pilot experiment at the Alvra end-station of *SwissFEL*[45]. This XFEL is designed for simultaneous operation in both hard (1.77–12.4 keV) and soft (180–1800 eV) X-ray regimes. We have used

the hard X-ray branch that also covers the 'tender' X-ray range 2–5 keV, giving access to photon energies not available at many other facilities. The accelerator is designed for operation up to an electron energy of 5.8 GeV and a repetition rate of 100 Hz. During the pilot phase, it was operated with a maximum electron energy of 2.3 GeV, which produced 200 μJ per pulse at 10 Hz repetition rate and a photon energy of 2.4 keV. The non-resonant X-ray emission experiment was performed using the full self-amplified spontaneous radiation (SASE) spectrum. The X-rays were focused to $20 \times 20 \ \mu m^2$ at the sample position using Kirkpatrick–Baez (KB) mirrors[58] with $B_4C/Mo$ coating.

X-ray emission spectra were acquired using the von Hamos geometry X-ray emission spectrometer at the Alvra Prime instrument. Single cylindrical Si(111) crystal with 1 mm segments and 25 cm focal radius dispersed the X-ray fluorescence with Bragg angles around 79 degrees[46,59]. Dispersed photons were registered on a per-shot basis by a 4.5 M JUNGFRAU detector[60,61] ($75 \times 75 \ \mu m^2$ pixel size and $4 \times 72 \ cm^2$ active area) which allows operation at any Bragg angle in the 40–80 degrees range without any detector motion. JUNGFRAU is a charge integrating pixel hybrid detector designed specifically for XFELs. It has automatic gain switching for each pixel, which makes it ideal for both photon-counting and high-intensity diffraction measurements. Due to the low photon flux on the detector, the JUNGFRAU was operated in high gain mode, where it allowed single photon detection at energies as low as 1.5 keV. Since the detector was used for the first time, it was running at 20 Hz repetition rate so that dark images for the best calibration (pedestal subtraction) could be collected throughout the measurement. Photoexcitation has been performed using a laser system based on Ti:sapphire amplifier (Legend Elite Duo HE+) combined with Vitara oscillator, Revolution pump lasers, and optical parametric amplifier HE-Topas Prime with NirUVis module[62]. The laser fluence of ~160 mJ cm$^{-2}$ has been achieved at the excitation wavelength of 450 nm with the pulse energy of 10 μJ, spot size at the sample position $80 \times 100 \ \mu m^2$ and the repetition rate 5 Hz.

Measurements were performed in a chamber filled with He to a pressure of 800 mbar. The chamber was evacuated periodically to remove any build-up of solvent vapor, which represents a significant loss in X-ray flux in this wavelength regime. The sample was recirculated in a closed cycle flow system by two HPLC pumps forming a round jet with 50–100 μm diameter in the chamber. The solution with $[Cu_4(PCP)_3]BAr^F_4$ concentration of 8.3 mM in THF was continuously purged with He and additionally cooled down to 10 °C to minimize the evaporation of the solvent to the chamber.

For the data reduction, each 3000 laser-on and laser-off XES spectra from individual X-ray pulses (forming one run) were summed. In total, we acquired data from ~366'000 couples of laser on/off X-ray pulses. Then principal component analysis was applied to the series of 122 runs corresponding to the delay between pump and probe pulses varied in the range 1.4 ns ($t_0$ + 900 ps, +500 ps, +100 ps, +5 ps, +1 ps, −100 , and −500 ps). It revealed only one statistically meaningful component, which can be assigned to the triplet excited state of $[Cu_4(PCP)_3]^+$. Acquisition of kinetic traces with many time points would require longer acquisition (or higher repetition rate of the experiment and higher flux) which was not possible during the pilot phase of SwissFEL operation. The relative X-ray energy was calculated based on the geometry of the spectrometer and detector pixel size. The maximum of the P $K\alpha_1$ line for the ground state spectrum was set at 2014.4 eV.

**Pump-probe X-ray scattering measurements and analysis**. Pump-probe X-ray scattering measurements were performed at beamline ID09 at the European Synchrotron Radiation Facility (ESRF, Grenoble, France)[63]. Approximately 100 ps-long X-ray pulses with center energy 14.75 keV and energy bandwidth of $\Delta E/E =$ 1.3% were produced using the U17 undulator and were monochromatized with Ru/$B_4C$ multilayer optics. X-ray pulses with a repetition rate 1 kHz were selected from the sequence of pulses generated by the storage ring operating in the 16-bunch mode using a combination of heat load and fast X-ray choppers[64]. The beam was focused to $40 \times 60 \ \mu m^2$ on the sample jet using toroidal X-ray mirror. Optical pump pulses with the center wavelength of 400 nm and 1 kHz repetition rate were produced by second harmonic generation (SHG) of the Ti:sapphire laser system. The size of the laser beam at the sample position was $250 \times 300 \ \mu m^2$ and the pulse energy was 140 μJ, which corresponds to the fluence of ~240 mJ cm$^{-2}$. The scattered X-ray photons were collected by an area detector (Rayonix MX170-HS, $1920 \times 1920$ pixels, 89 μm pixel size). Dissolved sample (~100 mL) with concentration 2.5 mM was circulated in the flow system and a round jet with the diameter 0.5 mm was formed by the quartz capillary nozzle. The jet was installed in a He-filled chamber and the solution in the sample reservoir was continuously purged with He to avoid interaction of $[Cu_4(PCP)_3]^+$ with oxygen.

X-ray scattering signals ($S(Q)_t$) for each time delay $t$ were obtained by azimuthal integration of 1000 individual detected 2D images with 3 s integration time. Normalization of images has been performed in the momentum transfer ($Q$) range 4.5–7.4 Å$^{-1}$. Difference signals $\Delta S(Q)_t$ were calculated by subtracting $S(Q)_{off}$ (acquired with the laser pulses arriving after the X-ray probe pulses) from $S(Q)_t$. The principal component analysis shows that $\Delta S(Q)_t$ can be well represented as a superposition of 3 components. Two of them are identical for all time delays including negative ones (Supplementary Fig. 3) and represent the fluctuation of the background while the third one represents the actual pump-probe signal. Individual $\Delta S(Q)_t$ curves were detected and removed as outliers if they deviated significantly (by more than 2 median absolute deviations) from a linear

combination of the 3 principal components. After removal of the PCA-identified background components (Supplementary Fig. 4) and adjustment of the sample detector distance by 4% the pump-probe signal for 1 ns delay was fitted as a linear combination of the solvent difference scattering signal arising from temperature changes (Supplementary Fig. 2), solvent signal arising from density change, dominating the signal at 2 μs delay, and the contribution from the solute, $\Delta S$ $(Q)_{theory}$, arising from the theoretically predicted $[Cu_4(PCP)_3]^+$ structural changes. The data analysis method is described in details in ref. [65]. The fitting was performed for the Q weighted difference signal in the Q range 0.5–6.5 Å$^{-1}$. $\Delta S$ $(Q)_{theory}$ was calculated via the Debye equation using DFT models of the ground- and excited-state structures[31] as the input.

**Sample preparation**. $[Cu_4(PCP)_3]BAr^F_4$ ($PCP = 2,6\text{-}(Ph_2P)_2C_6H_3^-$, $Ar^F = 3,5\text{-}(F_3C)_2C_6H_3$) was prepared following the previously published procedures[31]. All manipulations were performed under a protective atmosphere of dry nitrogen. In a representative sample preparation procedure 0.920 g, 0.375 mM of $[Cu_4(PCP)_3]$ $BAr^F_4$ were pre-weighted in a glovebox and stored in a Schlenk vessel capped with a Teflon valve for transportation. The bright green solid of $[Cu_4(PCP)_3]BAr^F_4$ was freshly dissolved in 150 mL of THF prior each experiment (tetrahydrofuran, 99.85%, extra dry, degassed, non-stabilized, AcroSeal®) to give a 2.5 mM solution using standard Schlenk-line techniques. The sample was subsequently transferred to the experimental set-up and rapidly injected via syringe under inert conditions into the sample container reservoir. The solution was maintained under inert atmosphere (helium or nitrogen was bubbled constantly through the solution) at all times during the measurement.

**DFT calculations**. DFT calculations of the electronic structure for ground state singlet and lowest triplet states were performed using Gaussian 09[66] and ADF-2018[57] packages. With Gaussian, the atomic charges were calculated using the Mulliken and NBO[67] approaches as well as Bader analysis[68,69] of the electron density topology. The electron density integration over atomic basis was carried out with the AIMALL package[70]. The well known B3LYP[71] and more recently developed M06[72] hybrid functionals were employed together with the DGDZVP[73,74] double-ζ and extended triple-ζ 6-311G(d,p)[75–77] basis set. Calculations using ADF with Slater-type atomic orbitals were also performed at various levels of theory. We used the triple-ζ basis set with two polarization functions (TZ2P) and quadruple-ζ basis set with four polarization functions (QZ4P). The gradient corrected GGA-PBE[78], meta-GGA TPSS[79], and hybrid B3LYP* and B3LYP[80] functionals were employed. Solvent effects of THF were simulated within the COSMO model[81]. Charges were calculated with Mulliken and Bader approaches. All calculations were performed with the same optimized geometries of the complex reported earlier[31].

## Data availability
Raw data were generated at the SLS, SwissFEL and ESRF large-scale facilities. Source data have been deposited at figshare and include measured data that have been used to obtain the results presented at Fig. 2: https://doi.org/10.6084/m9.figshare.11872347, Fig. 3: https://doi.org/10.6084/m9.figshare.11871756 and Fig. 4: https://doi.org/10.6084/m9.figshare.11872512.v1 Other data are available from the corresponding authors upon reasonable request.

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

## Acknowledgements

We acknowledge the ESRF for provision of beamtime at the ID09 beamline (proposal SC-4797), SLS for the beamtime at SuperXAS beamline and SwissFEL for the beamtime at Alvra. We thank L. Sala, C. Svetina, I. Usov, S. Ebner, S. Redford, D. Ozerov and R. Follath for the support during the SwissFEL experiment. K.H. acknowledges support from DANSCATT. A.G., Y.R., and A.T. acknowledge the financial support from Russian Foundation for Basic Research, project 18-02-40029. J.S. acknowledges the National Science Centre, Poland (NCN), for support under grants no. 2015/18/E/ ST3/00444 and 2015/19/B/ST2/00931. G.F.M. and C.B. acknowledge the support of the InterMUST Women Fellowship. S.K. and E.R. acknowledge the support of Russian Science Foundation, Project 18-13-00356. Support from NCCR MARVEL, NCCR MUST, and Energy System Integration (ESI) platform at PSI is also acknowledged. M.V. acknowledge support by the Central Research and Development Fund (CRDF) of the University of Bremen and generous financial support from the Fonds der Chemischen Industrie (FCI).

## Author contributions

G.S and M.V designed the study. G.S and N.A. performed pump-probe XAS experiment. G.S., J.S., C.C., G.K., R.B., S.M., G.P., D.G., M.B., D.J., C.B., and G.F.M. performed pump-probe XES experiment under the leadership of C.J.M. A.M. developed the data acquisition system based on JUNGFRAU detector, J.S., W.K., and J.C.-M. prepared and commissioned XES spectrometer. G.S., K.H., A.G., M.L., A.T., V.S., M.S., and V.K. performed pump-probe X-ray scattering experiment. G.S and A.G. performed calculations and interpretations of XAS. G.S. and K.H. performed calculation and interpretation of X-ray scattering data. G.S. and C.C. analyzed XES data. M.G. and A.C. performed preliminary optical characterization required for the design of experiments. S.K. and E.R. performed DFT calculation using Gaussian. A.G. performed DFT calculations using ADF. M.O., J.B., and M.V. synthesis and compound design, M.V. on-site sample preparation. G.S. has drafted the manuscript.

## Competing interests

The authors declare no competing interests.
