## [Peer Review File · Nature Communications]

Reviewers' comments:

Reviewer #1 (Remarks to the Author):

This is a very interesting manuscript that reports the results of time-resolved optical pump and x-ray probe measurements supported by calculations. I find this interesting for two reasons. First, the authors nicely combine novel methods, namely time-resolved x-ray absorption, x-ray emission and x-ray scattering measurements. Each was done at a different x-ray facility and this, I agree with the authors, nicely demonstrates the complementarity of the information contents and the facilities. Second, the study reports a nice combination of comparably new x-ray observables (excited-state information on electronic and geometric structures) on a sample for applications (material for OLED applications).

The main claims of the paper, accordingly, I think are: The first important claim is that this combination of methods is feasible and that it gives important information on state-of-the-art samples for applications in OLED technology. The second important claim is that the findings on how charge transfer between the metal centers in the material and the ligands interplay with structural changes may influence thinking in how to design such samples for improvements in their applications. Both claims are novel, I think. The results will be of broad interest because they bridge from new methodological aspects (time-resolved tender x-ray emission spectroscopy) to applications in a certain class of materials. By just being able to make these claims, I believe that this manuscript will influence thinking in the field (use of novel x-ray methods for samples in applied fields).

The data and analyses seem valid and the presentation and discussions are very clear and convincing.

I can therefore safely recommend publication of this manuscript in Nature Communications. Because some aspects are so new and because others are obscure to some extent I suggest considering the following comments that I expect to lead to a revised version of the manuscript.

Abstract

The second and third sentences seem completely unrelated. An additional sentence may help.

In “We found that excitation leads to delocalized charge movement from...” I do not understand what “delocalized charge movement” would be (as opposed to “localized charge movement” which I also would not know what it would be)? In addition, corresponding to the “from” I am missing the “to” in the claim. In the main text, when discussing charge transfer, this will also have to be addressed (charge is flowing from where to where?).

The sentence “The organometallic small-cluster...luminophores” is not accessible I think and should be simplified.

Main text

On p. 4, the three paragraphs (top on questions in research of luminophores, middle on x-ray methods and bottom on the specific sample studied here) seem unrelated. A few sentences on how this study relates the three aspects are needed as a start of what follows I think.

On p. 5 the sentence “When considering the details of...to the triplet state...” needs more information I think on how the concrete questions to be answered here are listed, which is very good, related in general to research on luminophores.

On p. 5 I suggest removing “modern” from the sentence “...we used a combination of modern time-resolved X-ray techniques...” because this is a very subjective term and it may be outdated in a few years.

In the section on the P Kalpha XES measurements:

On p. 7 I have problems with the sentence “Pump- probe X-ray emission spectroscopy is a technique requiring a high photon flux and efficient photoexcitation, a condition that can be more easily achieved at low repetition rates.” I believe I understand what the authors mean but I think that this can be very misleading. What is “low”, low compared to what? Do you mean low compared to the repetition rates at synchrotrons? Written as is I think that the statement suggests that 10 Hz is ideal

but I believe that higher rates would be even better (not MHz due to the lower photo excitation yield). I think that this potential misinterpretation should be avoided.

On p. 8 I find that "...spectrometer that disperses X-rays in the horizontal plane and focuses them vertically." seems to contradict Figure 3 (a) (rays between the crystal and the detector, probably hard to draw, but figure and text should be consistent).

On p. 8 in "...spectrometer allows for the full X-ray emission spectrum to be measured on a shot-to-shot basis, which is the most efficient strategy for X-ray sources with fluctuating X-ray intensity such as XFELs" I have a number of issues and some may be due to a misleading argumentation. I agree that this is an efficient method (due to multiplexing or however one wants to call it). I wonder what the authors mean with "normalization". Normalization to what? I zero? But then what would be different if they were not able to measure on a shot-by-shot basis and instead accumulated shots and averaged the signals? Wouldn't that still work for a spectrometers recording the whole spectrum? And wouldn't it also be possible to do that with a spectrometer recording only one energy (when averaging long enough, and here the authors seem to have accumulated data for several hours)? So what is the point (except for multiplexing)? What about advantages of shot-by-shot measurements such as being able to sort shots according to optical-x-ray arrival times or according to I zero (thereby avoiding too intense or too weak x-ray pulses)? A sentence on why the authors could not measure a kinetic trace (pump-probe signal as a function of delay time) would be helpful for the design of future experiments and to underline the heroic aspect of the reported data.

Caption of Figure 3 (c): At what time delay between optical pump and x-ray probe pulses was that measured?

On p. 9 for the interpretation of the XES shift in "From the comparison of the amplitude of transient XES with spectral changes induced by such shift, we can estimate ..." Do the authors assume a linear relationship between shift and charge transfer? For their estimate, what exactly did they do? I do not understand what "comparison of the amplitude of transient XES with spectral changes induced by such shift" really means here. Using a linear relationship with 0.1 eV corresponding to 1 electron charge, a charge of roughly 0.1 would correspond to a shift of 0.01 eV. The measured shift seems bigger so what is going on?

Further in the main text:

On p. 9 I find “X-ray pulses from the synchrotron converting a semi-continuous source to a pulsed one” quite confusing because a synchrotron, in general and with respect to this work in particular, clearly is a pulsed source and I do not see how it helps calling it “semi-continuous”.

On p. 11 I find the sentences “Avoiding the formation of a Cull center is of advantage because Cull prefers a square planar geometry and therefore is susceptible to the quenching by the solvent forming an exciplex^{38,52}, which requires significant energy for structural rearrangement.” and “At the same time, the rigid geometry reduces Huang–Rhys parameters that reduces the coupling of vibrational modes of the ground and excited states and decreases the probability of non-radiative losses.” not accessible. I think they will need to be simplified, broken up into several sentences and made of clearer statements.

On p. 12 the top paragraph is important. Can the authors try discussing this (strategies for making such materials) with more emphasize on the excited-state properties of the material, with more emphasize on how knowledge as gained in the present study would influence the design of such materials? Here, at the end of the paper, some speculation would be acceptable.

Reviewer #2 (Remarks to the Author):

This manuscript uses a combination of x-ray spectroscopy and scattering to measure the excited-state photophysics of a Cu₄ cluster relevant to OLED applications. The Introduction sets the stage very well, explaining the photophysics of temperature activated delayed fluorescence and the information accessible from x-ray photons. The three questions posed in the last paragraph of the Intro are well-chosen scientifically and also clearly orient the reader to the purpose of this work. This manuscript is also an introduction to the capabilities of the SwissFEL, and in particular its ability to measure x-ray emission spectroscopy in the “tender x-ray” region of the spectrum, in this case the emission from P atoms at ~2000 eV. All of this is measured on the ~100ps-to-us timescale.

The main scientific knowledge gained by this work is the observation that in the triplet excited state, both the C-coordinated and P-coordinated Cu atoms are partially oxidized, showing that the MLCT state is delocalized throughout the cluster. This is measured by peak shifts in the Cu K-edge XANES and in the P XES. X-ray scattering matches what would be expected from DFT calculations of the structural distortions in the triplet state.

My concern is whether the authors actually learned anything new – is this more than a technical demonstration of one beamline at the SwissFEL? They note when discussing the scattering that there is good DFT work that predicts the structure of the triplet state. Surely that calculation could also predict the charge shifts in the triplet, but there is no comparison of their experimental XANES/XES to that calculation. For example, their P XES gives them an estimate of a change of 0.097 electrons on each P: how does that compare to the Mulliken/Lowdin/etc charges in the singlet vs triplet state? What about the transient XANES – why not calculate the charge delocalization on the excited state and compare that to their experiment (or see if the FDMNES calculation of the excited state matches their transient spectrum)? As it is, I can't judge whether what they have done is confirm what was already suspected from theory or whether they have shown that the theory isn't good enough ("Experimental Verification" is one of their arguments for the development of these FEL sources). Their discussion says "Remarkably, our results demonstrate that the photexcited charge transfer does not come from only one Cu atom", but I don't see why that is so remarkable: the 4 Cus are bonded directly to each other. There are certainly mixed-valent multimetallic clusters with localized charges, but they normally have ligands between the metals. Perhaps there is good reason to have expected fully localized charges, but that claim needs to be justified.

If they do choose to make these comparisons in a revision, they will need to be more explicit about the photoexcitation conditions. In the XES section, they mention a ~70% excitation fraction – how does such a large fraction and the potential for multiphoton processes impact their scientific conclusion? Even that number was fuzzy: "we do not expect that it exceeds 70%". I didn't see the fraction for the other two experiments, but perhaps I missed it. This needs to be reported in mJ/cm², pump photons per molecule, or similar units (see recent work by Schlichting for the importance of multiphoton effects)

A few more technical comments:

1. In Figure 2, it would be useful to show vertical lines at the major transient features to make it easier for the reader to see the direction of the shift compared to the ground-state spectrum
2. For the scientific part of this, an explanation of why the P-coordinated and C-coordinated peaks in Fig 2 are different would be useful (what is the underlying electronic structure reason for the spectral difference)
3. On page 7, I was initially confused by the sentence starting with "Theoretical simulations of the ground state XANES". I finally figured out that the comma after "main maximum of the spectrum" needs to be removed. As is, it sounds like they are talking about the main maximum of the whole spectrum, not the main maximum of the [spectrum corresponding to P-coordinated sites]

Reviewers' comments:

Reviewer #1 (Remarks to the Author):

This is a very interesting manuscript that reports the results of time-resolved optical pump and x-ray probe measurements supported by calculations. I find this interesting for two reasons. First, the authors nicely combine novel methods, namely time-resolved x-ray absorption, x-ray emission and x-ray scattering measurements. Each was done at a different x-ray facility and this, I agree with the authors, nicely demonstrates the complementarity of the information contents and the facilities. Second, the study reports a nice combination of comparably new x-ray observables (excited-state information on electronic and geometric structures) on a sample for applications (material for OLED applications).

The main claims of the paper, accordingly, I think are: The first important claim is that this combination of methods is feasible and that it gives important information on state-of-the-art samples for applications in OLED technology. The second important claim is that the findings on how charge transfer between the metal centers in the material and the ligands interplay with structural changes may influence thinking in how to design such samples for improvements in their applications. Both claims are novel, I think. The results will be of broad interest because they bridge from new methodological aspects (time-resolved x-ray emission spectroscopy) to applications in a certain class of materials. By just being able to make these claims, I believe that this manuscript will influence thinking in the field (use of novel x-ray methods for samples in applied fields). The data and analyses seem valid and the presentation and discussions are very clear and convincing.

I can therefore safely recommend publication of this manuscript in Nature Communications. Because some aspects are so new and because others are obscure to some extent I suggest considering the following comments that I expect to lead to a revised version of the manuscript.

Abstract

The second and third sentences seem completely unrelated. An additional sentence may help.

We have added the sentence:

“Understanding and experimental verification of charge transfer in luminophores are needed for this development. Organometallic multicore Cu complex comprising Cu–C and Cu–P bonds, that we have studied, represents a new type of such luminophores. To investigate the charge transfer and structural rearrangements in this material, we applied three complementary pump-probe X-ray techniques...”

In “We found that excitation leads to delocalized charge movement from...” I do not understand what “delocalized charge movement” would be (as opposed to “localized charge movement” which I also would not know what it would be)? In addition, corresponding to the “from” I am missing the “to” in the claim. In the main text, when discussing charge transfer, this will also have to be addressed (charge is flowing from where to where?).

We have removed the word “delocalized”. From the context it is clear that the charge that was moved is distributed between a few atoms (and that was the reason why we called it

delocalized). The charge moves “to phenyl rings”, which is mentioned in the new version of the sentence:

“We found that the excitation leads to charge movement from C- and P- coordinated Cu sites and from the phosphorus atoms to phenyl rings”

The sentence “The organometallic small-cluster...luminophores” is not accessible I think and should be simplified.

We have simplified the sentence:

“The use of a cluster of attracting Cu atoms bonded to the ligands through C and P atoms is a novel and efficient way to keep structural rigidity of luminophores.”

Main text

On p. 4, the three paragraphs (top on questions in research of luminophores, middle on x-ray methods and bottom on the specific sample studied here) seem unrelated. A few sentences on how this study relates the three aspects are needed as a start of what follows I think.

We have added the sentence that links the three aspects:

“In the present study we investigate a material exhibiting TADF, namely $[Cu_4(PCP)_3]^+$ shown in Fig 1a, using pump-probe X-ray methods.”

On p. 5 the sentence “When considering the details of...to the triplet state...” needs more information I think on how the concrete questions to be answered here are listed, which is very good, related in general to research on luminophores.

We have added the following sentences to relate the question that we have investigated and general research on Cu luminophores:

“When considering the details of the electronic density and local structure changes of Cu-based TADF materials, the situation is clear for mononuclear complexes: MLCT transitions occur and for efficient complexes these electronic transitions are accompanied by minimal structural rearrangements around Cu. Such conclusions were supported by theory and summarized in recent reviews^{11,16}. For $[Cu_4(PCP)_3]^+$ only basic experimental data are available³¹ and the theory describing the mechanism of luminescence requires experimental verification. The organo-multicopper cluster motif is new, therefore, there is no well studied analogues and it is not obvious how to transfer knowledge about the mechanism of luminescence of mononuclear Cu complexes to the multinuclear case. Therefore...”

On p. 5 I suggest removing “modern” from the sentence “...we used a combination of modern time-resolved X-ray techniques...” because this is a very subjective term and it may be outdated in a few years.

It has been removed

In the section on the P Kalpha XES measurements:

On p. 7 I have problems with the sentence “Pump- probe X-ray emission spectroscopy is a technique requiring a high photon flux and efficient photoexcitation, a condition that can be more easily achieved at low repetition rates.” I believe I understand what the authors mean

but I think that this can be very misleading. What is “low”, low compared to what? Do you mean low compared to the repetition rates at synchrotrons? Written as is I think that the statement suggests that 10 Hz is ideal but I believe that higher rates would be even better (not MHz due to the lower photo excitation yield). I think that this potential misinterpretation should be avoided.

We have modified the sentence:

“Pump-probe X-ray emission spectroscopy is a technique requiring a high photon flux and efficient photoexcitation, conditions that can be more easily achieved at the repetition rate of XFELs rather than at synchrotrons.”

On p. 8 I find that “...spectrometer that disperses X-rays in the horizontal plane and focuses them vertically.” seems to contradict Figure 3 (a) (rays between the crystal and the detector, probably hard to draw, but figure and text should be consistent).

We believe that the figure and the text corresponds to each other. We have added an additional sentence for clarification:

“In this configuration the point of interaction of X-rays with the sample, cylindrical crystal and JUNGFRÄU detector are located in the same horizontal plane.”

On p. 8 in “...spectrometer allows for the full X-ray emission spectrum to be measured on a shot-to-shot basis, which is the most efficient strategy for X-ray sources with fluctuating X-ray intensity such as XFELs” I have a number of issues and some may be due to a misleading argumentation. I agree that this is an efficient method (due to multiplexing or however one wants to call it). I wonder what the authors mean with “normalization”. Normalization to what? I zero? But then what would be different if they were not able to measure on a shot-by-shot basis and instead accumulated shots and averaged the signals? Wouldn't that still work for a spectrometers recording the whole spectrum? And wouldn't it also be possible to do that with a spectrometer recording only one energy (when averaging long enough, and here the authors seem to have accumulated data for several hours)? So what is the point (except for multiplexing)? What about advantages of shot-by-shot measurements such as being able to sort shots according to optical-x-ray arrival times or according to I zero (thereby avoiding too intense or too weak x-ray pulses)? A sentence on why the authors could not measure a kinetic trace (pump-probe signal as a function of delay time) would be helpful for the design of future experiments and to underline the heroic aspect of the reported data.

The main point we wanted to make was that the most efficient way to collect data at an unstable, pulsed source is to collect a complete X-ray emission spectrum per pulse. This then allows any shot-to-shot data sorting to be applied with any desired filter (arrival time, intensity, pulse duration, photon energy etc.). An additional advantage is that for a measurement where nothing is being scanned, as in this case where non-resonant X-ray emission was recorded, the spectral shape is independent on the incident X-ray intensity (I_0) since every pulse represents the same experimental conditions and the data can be simply accumulated. This approach was taken to reduce complexity, as this was the first experiment at this instrument and at a facility that had only produced X-rays for the first time a few weeks before. The ability to normalize or sort data according to various additional diagnostics is certainly something that needs to be done for any type of scan, but for this measurement we deliberately avoided this requirement by employing a scan-free dispersive geometry. As the various detectors and data acquisition systems were being used for the first time we did not have the reliable capability of measuring shot-to-shot intensities, making

scanning any degree of freedom problematic. To obtain sufficient statistics to average out the XFEL fluctuations would have required significantly longer acquisitions.

To clarify the approach for taking data during these measurements we have updated the text of the paper to read:

“Non-resonant P K α XES spectra were measured using a von Hamos spectrometer that disperses X-rays in the horizontal plane and focuses them vertically. In this configuration the point of interaction of X-rays with the sample, cylindrical crystal and JUNGFRÄU detector are located in the same horizontal plane. This type of spectrometer allows for the full X-ray emission spectrum to be measured on a shot-to-shot basis, which is the most efficient strategy for X-ray sources with fluctuating X-ray intensity such as XFELs⁴⁶ since it collects a full spectrum for every XFEL pulse, allowing the spectra to be sorted and filtered on a shot-to-shot basis. For measurements such as this, where the data were collected non-resonantly with no scanning elements, the spectral shape of XES is independent on the incident X-ray intensity and the spectra can be simply summed together for efficient data collection”

We have added also the requested sentence about the measurements of kinetic traces in the fourth paragraph of the section “Methods”, “Pump-probe XES measurements and analysis”:
“Acquisition of kinetic traces with many time points would require longer acquisition (or higher repetition rate of the experiment and higher flux) which was not possible during pilot phase of SwissFEL operation.”

Caption of Figure 3 (c): At what time delay between optical pump and x-ray probe pulses was that measured?

The spectrum was extracted using Principal Component Analysis from the series of spectra corresponding to different delays (as described in the Methods section). To make sure that it is clear from the main text we have added the sentence:

“The transient has been extracted using principal component analysis from the measurements for delays varied in the range 1.4 ns.”

On p. 9 for the interpretation of the XES shift in “From the comparison of the amplitude of transient XES with spectral changes induced by such shift, we can estimate ...” Do the authors assume a linear relationship between shift and charge transfer? For their estimate, what exactly did they do? I do not understand what “comparison of the amplitude of transient XES with spectral changes induced by such shift” really means here. Using a linear relationship with 0.1 eV corresponding to 1 electron charge, a charge of roughly 0.1 would correspond to a shift of 0.01 eV. The measured shift seems bigger so what is going on?

Yes we have assumed a linear relationship between the shift of XES δE and charge changes at P atoms δq : $\delta E = A\delta q$ The validity of this approximation for XES spectra of P has been demonstrated in the literature (Petric, M. & Kavčič, M. J. Anal. At. Spectrom. **31**, 450 (2016); Petric, M. et al. Anal. Chem. **87**, 5632 (2015)) Approximate estimations of the reviewer are correct. Since the shift is very small it cannot be measured by comparing the position of maxima for laser-on and laser-off spectra. Instead one can use expansion of spectrum as a function of energy with one order of smallness: $XES(E, \delta E) = XES(E) + \frac{dXES}{dE}(E)\delta E$. Then the difference between excited and ground state spectra for a charge change δq can be calculated as $\Delta XES(E, \delta q) = \frac{dXES}{dE}(E)A\delta q$ The transient signal scales

down with the excited state fraction α : $TransientXES(E, \delta q) = \alpha \frac{dXES}{dE}(E)A\delta q$. Coefficient A is known from the measurements for reference compounds (Petric, M. & Kavčič, M. J. Anal. At. Spectrom. **31**, 450 (2016); Petric, M. et al. Anal. Chem. **87**, 5632 (2015)), TransientXES(E) and XES(E) we have measured experimentally. Therefore, if we can make a high estimate for the excited state fraction α , then the low estimate for the charge change δq can be obtained. We have added this description in the SI section “Charge change estimation from transient XES”.

Further in the main text:

On p. 9 I find “X-ray pulses from the synchrotron converting a semi-continuous source to a pulsed one” quite confusing because a synchrotron, in general and with respect to this work in particular, clearly is a pulsed source and I do not see how it helps calling it “semi-continuous”.

We have removed part of the sentence. The revised version is:

“In this setup, a fast mechanical X-ray chopper is used to isolate individual X-ray pulses from the synchrotron at a 1 kHz repetition rate”

On p. 11 I find the sentences “Avoiding the formation of a CuII center is of advantage because CuII prefers a square planar geometry and therefore is susceptible to the quenching by the solvent forming an exciplex^{38,52}, which requires significant energy for structural rearrangement.” and “At the same time, the rigid geometry reduces Huang–Rhys parameters that reduces the coupling of vibrational modes of the ground and excited states and decreases the probability of non-radiative losses.” not accessible. I think they will need to be simplified, broken up into several sentences and made of clearer statements.

We have simplified these sentences:

“In the excited state, it is advantageous to prevent the formation of a four-coordinate Cu^{II} with relatively flexible coordination sphere. Cu^{II} prefers a square planar geometry. Therefore, ligands often rearranges in such excited states so that the metal interacts with the solvent. This effect is known as exciplex formation and was observed previously^{38,52}. It leads to the quenching of the excited state because significant energy is required for the structural rearrangement.”

and

“The probability of non-radiative losses depends on the coupling of vibrational modes of the ground and excited states, which can be described using Huang–Rhys parameters. For rigid complexes that demonstrate small structural difference between ground and excited state Huang–Rhys parameters are minimal. Our data demonstrate that such a situation occurs for [Cu₄(PCP)₃]⁺.”

On p. 12 the top paragraph is important. Can the authors try discussing this (strategies for making such materials) with more emphasize on the excited-state properties of the material, with more emphasize on how knowledge as gained in the present study would influence the design of such materials? Here, at the end of the paper, some speculation would be acceptable.

We have added a few sentences in this paragraph:

“Further development of such complexes can be based on the alteration of the P-donors and phenyl-backbone to tune the emission maximum. Since both Cu and P atoms are involved in

the charge transfer in the excited state it is worth to consider modifications of ligands to change independently the electronic density at P and Cu centers and to explore their influence on the excited state and luminescence properties. Computational methods with experimentally verified parameters can be of great help predicting optimal ligands that correspond to the most efficient TADF emitters.”

Reviewer #2 (Remarks to the Author):

This manuscript uses a combination of x-ray spectroscopy and scattering to measure the excited-state photophysics of a Cu₄ cluster relevant to OLED applications. The Introduction sets the stage very well, explaining the photophysics of temperature activated delayed fluorescence and the information accessible from x-ray photons. The three questions posed in the last paragraph of the Intro are well-chosen scientifically and also clearly orient the reader to the purpose of this work. This manuscript is also an introduction to the capabilities of the SwissFEL, and in particular its ability to measure x-ray emission spectroscopy in the “tender x-ray” region of the spectrum, in this case the emission from P atoms at ~2000 eV. All of this is measured on the ~100ps-to-us timescale.

The main scientific knowledge gained by this work is the observation that in the triplet excited state, both the C-coordinated and P-coordinated Cu atoms are partially oxidized, showing that the MLCT state is delocalized throughout the cluster. This is measured by peak shifts in the Cu K-edge XANES and in the P XES. X-ray scattering matches what would be expected from DFT calculations of the structural distortions in the triplet state. My concern is whether the authors actually learned anything new – is this more than a technical demonstration of one beamline at the SwissFEL? They note when discussing the scattering that there is good DFT work that predicts the structure of the triplet state. Surely that calculation could also predict the charge shifts in the triplet, but there is no comparison of their experimental XANES/XES to that calculation. For example, their P XES gives them an estimate of a change of 0.097 electrons on each P: how does that compare to the Mulliken/Lowdin/etc charges in the singlet vs triplet state? What about the transient XANES – why not calculate the charge delocalization on the excited state and compare that to their experiment (or see if the FDMNES calculation of the excited state matches their transient spectrum)? As it is, I can’t judge whether what they have done is confirm what was already suspected from theory or whether they have shown that the theory isn’t good enough (“Experimental Verification” is one of their arguments for the development of these FEL sources).

To demonstrate that the experimental verification of computational models is necessary and useful and that our experimental data provide new relevant insights about the charge shift in the excited state we have performed a series of DFT calculations to estimate atomic charges with different methods including the Mulliken approach. We have added the comparison with our charge estimation from XES and qualitative results about the charge shifts from XANES. This is reported in Table 1 and in the SI Tables S1, S2 and S3. The following new paragraph has been added to the Discussion section:

“To compare our experimental results about charge shifts with theoretical data we have summarized in Table 1 a few representative examples of DFT calculations obtained using the Gaussian and ADF codes. Average changes of the charges are given in this table for P atoms, C-coordinated Cu and P-coordinated Cu atoms. An extended version of this table, as well as individual charges for all Cu and P atoms calculated using different methods, are

provided in the SI. From Table 1 one can give a broad range of interpretations. From the NBO charges obtained with Gaussian at the B3LYP/6-311G(d,p) level of theory one can conclude that the electron transfer occurs only from C- coordinated Cu sites. Using the same code, but with a smaller basis set (B3LYP/DGDZVP) one can deduce that the negative charge moves from the C-coordinated to P-coordinated Cu atoms. ADF at TPSS/QZ4P level of theory gives an opposite result: the electron density moves from the P-coordinated to C-coordinated Cu atoms. Using ADF (B3LYP/ QZ4P) and Mulliken charges we see the shift of the electronic density from both C coordinated and P- coordinated Cu atoms, which is in agreement with the qualitative picture obtained from the pump-probe XANES. The Bader charges computed at the B3LYP/TZ2P level of theory testify for a significant contribution of P atoms in addition to the C-coordinated Cu atoms. In the set of calculations that we have performed we have not found the approach that shows the electronic density shift from both types of Cu and P atoms. The charge change at P atoms (0.020 e) obtained using ADF (B3LYP*/TZ2P) is in agreement with our experiment-based low estimate for the DFT charge change (0.013 e). Thus, a broad range of theories about charge shift in $[\text{Cu}_4(\text{PCP})_3]^+$ as a result of photoexcitation demonstrates the need of experimental verification based on pump-probe X-ray measurements.”*

Calculations using FDMNES are non self-consistent. Charge transfer between Cu atoms and ligands is a free parameter. Therefore, they do not generate a charge transfer model that can be verified experimentally. While FDMNES works sufficiently well to reproduce the spectral changes related with different Cu coordination (C versus P) or originating from significant structural rearrangements in mononuclear metal complexes (see, for example, *J. Phys. Chem. A* **2008**, *112*, 5363) such simple approach does not allow to obtain adequate changes of charge for multicore Cu complex and subsequently to reproduce the transient XANES. Due to much higher popularity of DFT-based calculations in comparison to FDMNES calculations among Nature Communications readers, we decided to include only the discussion of DFT results in the manuscript.

Regarding the comparison of charge changes at P atoms from the experiment and DFT, it is important to note that the shift of emission lines can be calibrated to the formal charge (such calibration we have used in the previous version of the manuscript) or to the charge estimated using DFT (see Fig 2 of Petric et al *Anal. Chem.* 2015, *87*, 5632–5639). Formal charges are larger by almost a factor of 8. We have added such second calibration and corresponding clarification in the manuscript in the section “Probing the involvement of the phosphine ligands in the charge transfer”:

“From measurements on reference compounds,⁴⁷ the expected shift of the main line is 0.1 eV per 1.0 electron change of the formal P charge (oxidation state). Alternatively, one can use calibration to the DFT charge, for example calculated using Mulliken approach. In this case, the same 0.1 eV shift of the emission line corresponds to the 0.13 electrons of the DFT charge variation.”

Their discussion says “Remarkably, our results demonstrate that the photexcited charge transfer does not come from only one Cu atom”, but I don’t see why that is so remarkable: the 4 Cus are bonded directly to each other. There are certainly mixed-valent multimetallic clusters with localized charges, but they normally have ligands between the metals. Perhaps there is good reason to have expected fully localized charges, but that that claim needs to be justified.

In the initial version of the manuscript, we made this comment to stress that our experimental results demonstrate a picture that is different from DFT calculations reported in ref. 31. In the revised manuscript, that describes the broad range of results that can be obtained using DFT, the sentence has been revised:

“Some computational models, (for example, Mulliken charges calculated using Gaussian within B3LYP/6-311G(d,p) approach), shows that the charge moves mainly from one C-coordinated Cu atom. Nevertheless our experimental results demonstrate the involvement of two types of Cu centers”

If they do choose to make these comparisons in a revision, they will need to be more explicit about the photoexcitation conditions. In the XES section, they mention a ~70% excitation fraction – how does such a large fraction and the potential for multiphoton processes impact their scientific conclusion? Even that number was fuzzy: “we do not expect that it exceeds 70%”. I didn’t see the fraction for the other two experiments, but perhaps I missed it. This needs to be reported in mJ/cm², pump photons per molecule, or similar units (see recent work by Schlichting for the importance of multiphoton effects)

We have added the required information about photoexcitation conditions (in mJ/cm²) in the “Methods” section:

“Pump-probe XANES measurements and analysis”, first paragraph:

“The laser beam fluence of ~130 mJ/cm² was achieved by focusing to the 200-200 μm² spot at the sample position.”

“Pump-probe XES measurements and analysis“, second paragraph:

“The laser fluence of ~160 mJ/cm² has been achieved at the excitation wavelength of 450 nm with the pulse energy of 10 μJ, spot size at the sample position 80-100 μm² and the repetition rate 5 Hz.”

“Pump-probe X-ray scattering measurements and analysis”, first paragraph:

“The size of the laser beam at the sample position was 250-300 μm² and the pulse energy was 140 μJ, which corresponds to the fluence of ~240 mJ/cm².”

We did not use the values of excited-state fraction in the analysis: we analyzed only the shape of transient XANES and X-ray scattering spectra (and not the amplitude, which scales proportionally to the excited-state fraction). We used the amplitude of the transient XES spectrum to make the low estimate of charge changes. For that, we only need an upper-bound estimate (70%) for the excited state fraction and not the exact value of the excited state fraction.

With the target theoretical value of around 1 for the ratio between the number of absorbed photons and the number of molecules in the excited volume, we tried to maximize the pump-probe signal. Such a regime is typically used by many groups for pump-probe XAS/XES or X-ray scattering experiments. If we compare our excitation conditions with those from the recent article of Schlichting (Nature Communications 10 (2019) 3177), they have used 40 photons per retinal at the high excitation density regime in which clear changes of kinetics were observed. Our conditions are still far from such a high excitation density regime. Moreover, for the pump-probe XANES experiment, we used nanosecond excitation, which additionally reduces the peak power and any nonlinear absorption effects. Thus, a minor fraction of doubly excited molecules cannot be fully excluded in our experiments, but the percentage of such molecules is low in comparison to the fraction of molecules excited by a

single photon and therefore this effect cannot influence any conclusions reported in the manuscript.

A few more technical comments:

1. In Figure 2, it would be useful to show vertical lines at the major transient features to make it easier for the reader to see the direction of the shift compared to the ground-state spectrum

It has been implemented.

2. For the scientific part of this, an explanation of why the P-coordinated and C-coordinated peaks in Fig 2 are different would be useful (what is the underlying electronic structure reason for the spectral difference)

We have the following explanation:

“Qualitatively, such shift is due to the influence of the first coordination sphere of Cu: for the quasi-linear coordination by two carbon atoms, the orbitals contributing to the main XANES maximum are non-bonding p-orbitals of Cu, while the coordination by three P atoms results in the hybridization of the P and Cu p-orbitals which shifts the main XANES maximum to lower energies.”

3. On page 7, I was initially confused by the sentence starting with “Theoretical simulations of the ground state XANES”. I finally figured out that the comma after “main maximum of the spectrum” needs to be removed. As is, it sounds like they are talking about the main maximum of the whole spectrum, not the main maximum of the [spectrum corresponding to P-coordinated sites]

It has been corrected.

Additionally, to use consistently ADF in the whole manuscript we have changed the theoretical spectra presented at Fig 2 with calculations performed using ADF with computational parameters identical to those used in the new part about charge calculations. Spectra are similar to those reported in the previous version of the manuscript, which were actually also calculated using ADF. All the interpretation of spectra remains the same, only section Methods, “Pump-probe XANES measurements and analysis” has been revised accordingly:

“Theoretical XANES spectra have been obtained by calculating the probabilities of transitions between the core and virtual molecular orbitals using the approach described previously^{58,59}. Molecular orbitals were obtained by DFT using ADF code⁶⁰. Non-relativistic self-consistent calculations have been performed using the quadruple- ζ basis set with four polarization functions (QZ4P) and the hybrid B3LYP exchange-correlation functional.”

We have added also section “DFT calculations” that describes methods used to calculate the charges:

“DFT calculations of the electronic structure for ground state singlet and lowest triplet states were performed using Gaussian 09⁶⁹ and ADF-2018⁶⁰ packages. With Gaussian, the atomic charges were calculated using the Mulliken and NBO⁵³ approaches as well as Bader analysis^{54,55} of the electron density topology. The electron density integration over atomic basis was carried out with the AIMALL package⁷⁰. The well known B3LYP⁷¹ and more recently developed M06⁷² hybrid functionals were employed together with the DGDZVP^{73,74} double- ζ and extended triple- ζ 6-311 G(d,p)⁷⁵⁻⁷⁷ basis sets. Calculations using ADF with Slater-type atomic orbitals were also performed at various levels of theory. We used the triple- ζ basis set with two polarization functions (TZ2P) and quadruple- ζ basis set with four polarization functions (QZ4P). The gradient corrected GGA-PBE⁷⁸, meta-GGA TPSS⁷⁹ and

hybrid B3LYP* and B3LYP⁸⁰ functionals were employed. Solvent effects of THF were simulated within the COSMO model⁸¹. Charges were calculated with Mulliken and Bader approaches. All calculations were performed with the same optimized geometries of the complex reported earlier³¹.”

We have added the section “Data availability”:

“Raw data were generated at the SLS, SwissFEL and ESRF large-scale facilities. Source data have been deposited at figshare and include measured data that have been used to obtain the results presented at Figure 2: <https://doi.org/10.6084/m9.figshare.11872347>, Figure 3: <https://doi.org/10.6084/m9.figshare.11871756> and Figure 4 : <https://doi.org/10.6084/m9.figshare.11872512.v1> Other data are available from the corresponding authors upon reasonable request.”

REVIEWERS' COMMENTS:

Reviewer #2 (Remarks to the Author):

I appreciate the changes made by the authors, which have satisfied my concerns about the scientific impact of this work. I now support its publication. Only two new minor things to fix:

1) In the new abstract, I don't know what "attracting Cu atoms" means (specifically the word "attracting"). This should be rewritten.

2) In the responses to Rev 1, they added the phrase "conditions that can be more easily achieved at the repetition rate of XFELS rather than at synchrotrons" This is again ambiguous, with the new generation of MHz FELs coming online. Please be more clear exactly what you mean here.

Below is the copy of the reviewers' comments (black font) together with our point-by-point responses (blue font).

REVIEWERS' COMMENTS:

Reviewer #2 (Remarks to the Author):

I appreciate the changes made by the authors, which have satisfied my concerns about the scientific impact of this work. I now support its publication. Only two new minor things to fix:

1) In the new abstract, I don't know what "attracting Cu atoms" means (specifically the word "attracting"). This should be rewritten.

We have removed the phrase "attracting Cu atoms". The new version of the sentence is:

“The use of Cu cluster bonded to the ligands through C and P atoms is an efficient way to keep structural rigidity of luminophores.”

2) In the responses to Rev 1, they added the phrase "conditions that can be more easily achieved at the repetition rate of XFELs rather than at synchrotrons" This is again ambiguous, with the new generation of MHz FELs coming online. Please be more clear exactly what you mean here.

We have added “typically, below 1 kHz” to clarify this question:

“Pump-probe X-ray emission spectroscopy is a technique requiring a high photon flux and efficient photoexcitation, conditions that can be more easily achieved at the repetition rate of X-ray Free Electron Lasers, XFELs (typically, below 1 kHz) rather than at synchrotrons.”